# Community assessment of crustose calcifying red algae as coral recruitment substrates

**Mari Deinhart**[1]*, **Matthew S. Mills**[1,2], **Tom Schils**[1]

**1** Marine Laboratory, University of Guam, UOG Station, Mangilao, Guam, **2** School of Science, Technology, and Engineering, University of the Sunshine Coast, Sippy Downs, Queensland, Australia

* marideinhart@gmail.com

**Data Availability Statement:** All newly generated DNA sequences have been submitted to GenBank and their accession numbers are listed in the supplementary tables.

## Abstract

Successful recruitment of invertebrate larvae to reef substrates is essential to the health of tropical coral reef ecosystems and to their capacity to recover from disturbances. Crustose calcifying red algae (CCRA) are a species rich group of seaweeds that have been identified as important recruitment substrates for scleractinian corals. Most studies on the settlement preference of coral larvae on CCRA use morphological species identifications that can lead to unreliable species identification and do not allow for examining species-specific interactions between coral larvae and CCRA. Accurate identifications of CCRA species is important for coral reef restoration and management to assess CCRA community composition and to detect CCRA species that are favored as coral recruitment substrates. In this study, DNA sequence analysis, was used to identify CCRA species to (1) investigate the species richness and community composition of CCRA on experimental coral recruitment tiles and (2) assess if the coral *Acropora surculosa* preferred any of these CCRA species as recruitment substrates. The CCRA community assemblages on the coral recruitment tiles was species-rich, comprising 27 distinct CCRA species of the orders Corallinales and Peyssonneliales which constitute new species records for Guam. Lithophylloideae sp. 1 (Corallinales) was the CCRA species that was significantly favored by coral larvae as a recruitment substrate. Lithophylloideae sp. 1 showed to hold a valuable ecological role for coral larval recruitment preference. Lithophylloideae sp. 1 had the highest benthic cover on the recruitment tiles and contained most *A. surculosa* recruits. DNA barcoding revealed a high taxonomic diversity of CCRA species on a microhabitat scale and provided detailed insight into the species-specific ecological interactions between CCRA and corals. With a steady decline in coral cover, detailed information on species interactions that drive reef recovery is valuable for the planning of marine management actions and restoration efforts.

## Introduction

Coral reefs are threatened worldwide and are undergoing rapid change because of the increased frequency of coral bleaching events, eutrophication, sedimentation, and regime shifts [1–5]. Apparent declines in live coral cover are generally the first observed warning signs

**Funding:** This research is based upon work supported by the National Aeronautics and Space Administration (NASA; nasa.gov) and the National Science Foundation (NSF; nsf.gov) under grant numbers 80NSSC17M0052 and OIA-1946352 awarded to TS and managed through the Guam EPSCoR offices of NASA and NSF. Any opinions, findings, conclusions or recommendations expressed in this manuscript are those of the authors and do not necessarily reflect the views of NASA and NSF or any of their subagencies. The funders had no role in the study design, data collection and analysis, decision to publish, or preparation of the manuscript.

**Competing interests:** The authors have declared that no competing interests exist.

for changes in benthic composition on coral reefs. Many other organisms are intrinsic components of these ecosystems but have received less scientific attention than scleractinian corals. Crustose calcifying red algae (CCRA; representatives of the red algal orders Corallinales, Corallinapetrales, Hapalidiales, Sporolithales, and Peyssonneliales) are a species rich group of macroalgae that are a dominant component of coral reefs. Island groups in the tropical Pacific show significant differences in the abundance of CCRA on their reefs [6], however patterns in CCRA species diversity between these island groups is still largely unknown. CCRA are essential components of healthy reef systems because of their ecological importance as (1) important calcium carbonate producers [7], (2) carbon sequesters while creating benthic (micro)habitats [7], (3) the preferred settlement substrates for many invertebrate larvae (including scleractinian corals) [8–10], (4) stabilizers of reef structures [11], and (5) suppressors of fleshy algal overgrowth [12–14]. Despite CCRA being a large component to the benthic community in Guam's tropical reefs, the diversity and functional ecology of CCRA are still largely unexplored. Through reef accretion, CCRA provide structural complexity and promote habitat diversity [15].

The coral reefs of Micronesia are known for their high diversity of acroporid corals in the shallow forereef zones [16]. Acroporids are a major group of reef builders due to their high benthic cover and fast growth rates [16, 17], thus creating habitat diversity for other reef organisms [18, 19]. Guam's acroporid corals have undergone extensive mortality in recent years, particularly in the forereef zone, due to repeated bleaching events caused by episodes of elevated seawater temperatures in 2013, 2014, 2016, extreme low tides in 2015, and high predation from the sea star, *Acanthaster planci* [1–5, 20]. Understanding the factors that drive reef recovery is important as the frequency, severity, and diversity of disturbances impacting coral reefs continue to increase [18]. The recovery of coral reefs depends on the regeneration of coral populations through the successful recruitment of coral larvae [17, 21–23].

The continuing decline in acroporid corals [5] on Guam's reefs has led to coral restoration efforts that primarily focus on rearing corals in nurseries for outplanting efforts (L. Raymundo, personal communication). Due to its overall resilience to disturbances, *Acropora surculosa* Dana (1846) has been one of the main scleractinian coral species used for restoration efforts in Guam [5]. *A. surculosa* is a corymbose acroporid that has been well studied in Guam due to its common occurrence on forereefs, its fast growth rate, and its overall value as a reef builder [17, 24]. Corals used for restoration efforts are obtained through fragmentation of source colonies or via sexual reproduction. Sexually produced coral transplants can enhance genetic variability and can generate high numbers of new colonies to be used for large-scale restoration efforts [25, 26]. *A. surculosa* is a hermaphroditic broadcast spawner, with spawning events occurring during the last quarter lunar cycles of July and August [27].

CCRA are well-known to serve as the preferred settlement substrate for coral larvae [8, 9, 13, 28–30]. Research has largely focused on the specific abiotic and biotic factors that facilitate successful scleractinian coral larval recruitment, settlement, growth, and fecundity [8, 11, 31]. The University of Guam Marine Laboratory has hosted *s*tudies describing various mechanistic pathways that facilitate *A. surculosa* and *Leptastrea purpurea* larval settlement on CCRA [32–34], but species-specific recruitment patterns of *A. surculosa* larvae have not yet been conducted. CCRA species can possess species-specific chemical fingerprints and microbiome communities [35] that could influence recruitment patterns, highlighting the need to correctly identify the CCRA species that are favored for larval recruitment in the Micronesian region.

Experimental studies commonly identify CCRA based on morphological features and often details on the identifications are not reported [36]. Investigations using DNA sequencing have revealed that morpho-anatomical identifications of CCRA are often inaccurate [36], because of the specialized techniques required [37–40] and the high degree of cryptic diversity in

CCRA [41–47]. Species identification of CCRA can be notoriously challenging due to their simple morphologies, convergent evolution, phenotypic plasticity, and the frequent absence of reproductive structures [39, 48]. Despite dispersal limitations, many CCRA species are reported incorrectly to have broad distribution ranges, which are largely based on a morphospecies concept [49, 50]. Due to these identification challenges, CCRA are often grouped into a single functional group in monitoring and ecological surveys, which does not recognize the species richness of CCRA and the different ecological roles that CCRA species serve in marine environments, including tropical reefs, temperate reefs, and rhodolith beds [50]. To address the challenges of CCRA species identification, DNA sequence analysis has proven to be an effective tool to investigate species diversity in CCRA [41, 47, 51].

This study used DNA sequencing to investigate the species composition of CCRA communities on coral recruitment tiles in environmental conditions similar to natural reefs with well-developed *Acropora surculosa* stands (Pago Bay, Guam). Following the characterization of these CCRA communities, an analysis of recruitment preference by *A. surculosa* larvae for CCRA species was conducted. We hypothesized that (1) a diversity assessment using DNA barcoding would reveal more CCRA taxa than a morpho-anatomical diversity assessment and (2) *A. surculosa* larvae prefer to recruit on a select number of CCRA taxa. Based on previous recruitment studies [9, 29, 30, 52, 53] we hypothesize that *A. surculosa* larvae will favor members of the subfamily Lithophylloideae (Corallinales, Rhodophyta) as recruitment substrates. This study aims to provide new insights on CCRA diversity and community composition, and the ecological roles fulfilled by CCRA species.

## Materials & methods

### Substrate categories on coral recruitment tiles

Twelve ceramic star-shaped coral recruitment tiles [54] were maintained in the flow-through seawater system at the University of Guam Marine Laboratory (UOGML) to recruit CCRA one month prior to the July 2018 *Acropora* spawning event. The star-shaped tiles allowed for accurate measurements of organism cover on each side of the tiles (excluding the bottom surface). The seawater flowing into the holding tank was drawn from the coral reef at Pago Bay on the east coast of Guam. All twelve coral recruitment tiles were conditioned in the same tank, resulting in similar growing conditions (e.g., temperature, salinity, and turbidity).

Prior to the 2018 spawning event, *A. surculosa* coral colonies were collected on July 1, 2018, from Pago Bay and Tanguisson in Guam and transferred to a separate holding tank. Coral collection permits for research and education were issued by Guam Division of Aquatic and Wildlife Resources (DAWR). *A. surculosa* is a morphologically distinct coral species on Guam's reefs. Colonies from Tanguisson spawned first on July 7, 2018, and colonies from Pago Bay spawned on July 9, 2018. On the nights of the spawning event, released gametes, in the form of egg-sperm bundles, were collected from the water surface of the tank. Once gametes were collected, coral colonies were returned to the reef and reattached to minimize damage. After fertilization, embryos were kept in an experimental setting to develop into planula larvae. The larvae were then released into tanks with CCRA-covered tiles for settlement and recruitment.

Following the 2018 *A. surculosa* spawning event, the coral recruitment tiles were maintained in the holding tank for coral larval recruitment for 128 days, when photographs of each tile and all 11 exposed sides were taken of each tile on 13 December 2018. At the same time, scrapings of live CCRA tissue samples were taken for DNA extraction. Coral recruits were documented while taking photos of the tiles. All CCRA on which coral recruits were detected were sampled for DNA extraction. In addition, 32 samples of CCRA that were not associated with

**Table 1. Description of the nine substrate categories found on the recruitment tiles.** N Recruits, number of coral recruits found on each substrate category; N CCRA spp., number of CCRA species included in each substrate category; Morphological Characteristics, characteristic features for each substrate category. FP: fluorescence photography; VLP, visible light photography.

| Substrate Category | N Recruits | N CCRA spp. | Morphological Characteristics |
|---|---|---|---|
| Corallinales spp. | 3 | 13 | Bright pink/red in FP. Light pink to light purple in VLP. Conceptacles prevalent but smaller than the conceptacles found on Lithophylloideae sp. 1. |
| Lithophylloideae sp. 1 | 45 | 1 | Salmon red color in VLP. Bright orange in FP. Noticeably large conceptacles. |
| Lithophylloideae spp. 2–4 | 5 | 3 | Magenta in color in VLP. Smaller conceptacles than Lithophylloideae sp. 1. More conceptacles covering the surface area. Deep shade of orange or highlighter pink in FP. |
| Peyssonneliales spp. 1,3,6 | 0 | 3 | Deep red to dark brown color in VLP. Thin, smooth, and firmly attached thallus. |
| Peyssonneliales sp. 2 | 0 | 1 | Thallus is more fleshy, gelatinous, and thick than other Peyssonneliales spp. Bright red in VLP. Bright, light orange color in FP. |
| Peyssonneliales spp. 4,7 | 3 | 2 | Dark red thalli with yellow-olive margins in VLP. Highest variation in color shade and color range compared to other Peyssonneliales categories. |
| Peyssonneliales sp. 5 | 0 | 1 | Lighter shade of red than other Peyssonneliales spp. Thallus appears to be thinner and more firmly attached than other Peyssonneliales spp. |
| *Polystrata* spp. 1–3 | 4 | 3 | Calcified encrusting thallus that is firmly adherent to the substrate. Thallus can be layered. Red in VLP. |
| Bare substrate | 0 | 0 | Bleached or bare substrate with no CCRA or coral growth. |

coral recruits were also collected. Samples without associated coral recruits were chosen based on their unique morphology and replicate samples of these morphotypes were taken to address cryptic diversity. Following DNA extraction, tiles were air-dried and deposited into the marine plant collection of the University of Guam Herbarium (GUAM).

The recruitment preference of the coral *A. surculosa* was examined by first grouping CCRA taxa into eight substrate categories that could be discerned visually and a bare tile category (Table 1). The eight substrate categories included CCRA thalli that were not sequenced but had similar morphological characteristics to CCRA species validated through DNA sequencing analysis. Coral recruits were visually counted during the sampling process and were confirmed during the image analysis. During the image analysis, four small coral recruits were discovered to have been overlooked during the CCRA sampling process. The CCRA species to which these corals had recruited (*i.e.*, Corallinales spp. and Lithophylloideae sp. 1) were confidently identified in the digital images and were added to the dataset prior to further analysis. Substrate cover was measured using Adobe Photoshop version CC 2020 software. Color overlays were used to obtain the pixel count for each substrate category present on tile sides. The percent cover for each category was calculated by dividing the total number of pixels of each color (each representing one of the nine categories) by the total pixel count of the tile side (Fig 1). Pixels of coral recruits were ascribed to the substrate category on which they recruited.

## DNA barcoding

A total of 92 CCRA specimens were identified using DNA sequencing. For each specimen, a patch of thallus free from epiphytes was swabbed clean with a 10% bleach solution. A Dremel rotary tool, a pair of tweezers, or single-edged razor blades were used to scrape fragments from each specimen for DNA extraction. The Dremel, tweezers, and razor blades were sterilized by soaking them in 10% bleach and heating them over a flame after each tissue removal to avoid contamination. DNA of each CCRA specimen was extracted using QIAGEN DNeasy Blood & Tissue Kit (Qiagen Inc., Valencia, CA) or the GenCatch Blood & Tissue Genomic Mini Prep Kit (Epoch Life Science Inc., Missouri City, TX) following the manufacturer's bench protocol.

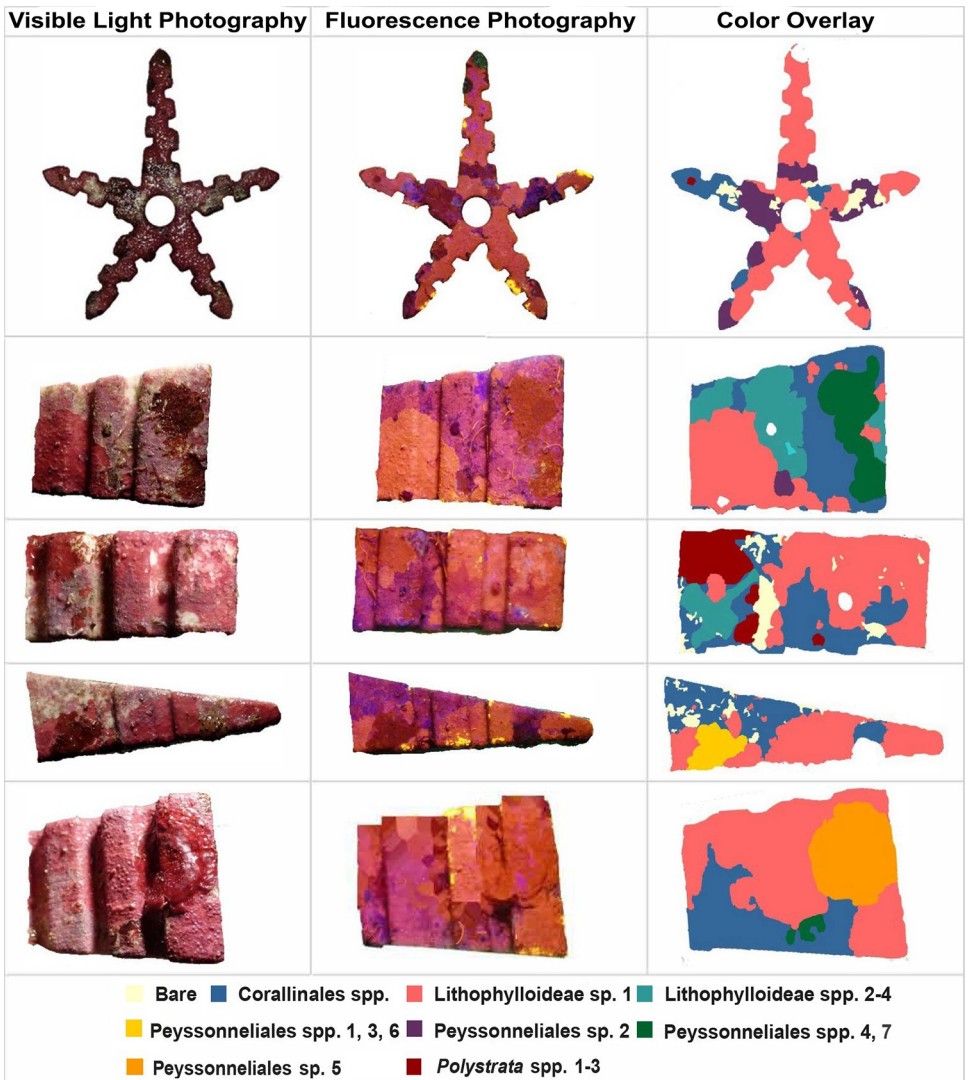

| Visible Light Photography | Fluorescence Photography | Color Overlay |

Legend:
■ Bare ■ Corallinales spp. ■ Lithophylloideae sp. 1 ■ Lithophylloideae spp. 2-4
■ Peyssonneliales spp. 1, 3, 6 ■ Peyssonneliales sp. 2 ■ Peyssonneliales spp. 4, 7
■ Peyssonneliales sp. 5 ■ *Polystrata* spp. 1-3

**Fig 1. Example of coral recruitment tile photographs under visible light and actinic light plus the color overlay used for the calculation of percent cover for each substrate category.** Benthic cover was assessed for all 11 exposed sides of the 12 tiles. Substrate category colors are the same as in Figs 4 and 5.

Three genetic markers were amplified via polymerase chain reaction (PCR) for species delimitation and identification. The mitochondrial cytochrome c oxidase subunit I, COI-5P (roughly 664 bp), is a barcode marker that is regularly used for DNA barcoding of red algae [55] and was used for both the Corallinales and Peyssonneliales. The primer combination used to amplify COI-5P was TS_COI_F01_10 (5′-TCGARTCYCGTCTCTCTCG-3′) [56] and the reverse primer, GWSRx [57]. Protocols for COI-5P amplification follow Mills & Schils [56].

Chloroplast photosystem II thylakoid membrane protein D1, *psb*A (roughly 950 bp), was used as a second barcode marker for the Corallinales. P*sb*A is more conserved than COI-5P and is frequently used for CCRA identification because of its high success rate of amplification [58]. The primers used to amplify this gene were psbAF and psbAR2 [59]. Amplification of *psb*A followed the PCR protocol in Mills & Schils [56]. The chloroplast ribulose-1, 5-biphosphate carboxylase large subunit, *rbc*L (roughly 1,350 bp), was amplified for a subset of

Corallinales specimens from the coral recruitment tiles. Amplification of *rbc*L used the primers F57 and rbcLrevNEW [60] following the PCR protocol outlined in Saunders and Moore [60].

## Species delimitation and phylogenetic analysis

PCR products were sent to Macrogen Inc. (Seoul, Republic of Korea) for DNA sequencing. Once chromatograms were obtained, consensus sequences were assembled using Geneious Pro 11.0.5 computer software (https://www.geneious.com; [61]). BLAST searches (Basic local alignment search tool) [62] of the consensus sequences were run to search for close matches in (1) a database of CCRA sequences from Guam and (2) online repositories such as GenBank and the Barcode of Life Database (BOLD) [63]. Gene alignments were created using the MUS-CLE plugin [64] in Geneious Pro 11.0.5. An alignment of COI-5P was made for the Peyssonne-liales and a concatenated alignment of COI-5P, *psb*A, and *rbc*L for the Corallinales. To assess species richness and to delimit putative species in this study, sequence divergence thresholds were calculated using the Automatic Barcode Gap Discovery (ABGD) [65] for COI-5P (3%), *psb*A (2.5%), and *rbc*L (0.9%) [55].

To resolve the taxonomic identity of CCRA, all sequences from the coral recruitment tiles were selected for phylogenetic analysis. Maximum likelihood (ML) was used to infer phylogenies through IQ-TREE 2 [66]. IQ-TREE 2 uses a combination of hill-climbing approaches and stochastic nearest neighbor interchange (NNI) operations to obtain higher likelihoods while estimating maximum likelihood phylogenies [66]. The concatenated alignment was partitioned by gene and the optimum evolutionary model for each gene was found using ModelFinder [67]. The Ultrafast Bootstrap Approximation was used to achieve unbiased node support values with 1000 replicates [68].

BLAST searches [62] of the Corallinales specimens did not reveal close matches (>97% sequence similarity) with publicly available DNA sequences. Therefore, taxonomic identifications of Corallinales specimens were derived from a phylogeny based on the seven-gene concatenated alignment of Peña et al. [69] (Fig 2), with the addition of the DNA sequences from the recruitment tiles. This alignment of Corallinophycidae (S1, S2 Tables; Fig 2) consisted of seven genes (23S rRNA, COI, EF2, LSU rRNA, *psb*A, *rbc*L, and SSU rRNA) and the total length of the seven-gene concatenated alignment was 11,608 bp. The length of the individual gene alignments was: 370 bp for 23S rRNA, 687 bp for COI, 1,622 bp for EF2, 4,716 bp for LSU, 784 bp for *psb*A, 1,386 bp for *rbc*L, and 2,086 bp for SSU. ModelFinder [67] in IQ-TREE 2 [66] found that the best-fit models for each partition were: TVMe+I+G4 (23S rRNA), TN+F+I+G4 (EF2), TIM3+F+I+G4 (LSU), and GTR+F+I+G4 (COI, *psb*A, *rbc*L, and SSU). The concatenated gene matrix of this alignment is incompletely filled, certain taxa are represented by gene sequences from more than one specimen (sometimes from different geographical areas), and some identifications might require nomenclatural updates as CCRA taxonomy is continuously being refined. Yet, we honor the expert opinion of Peña et al. [69] in generating this alignment and the taxonomic identifications that they settled on, particularly because the alignment only serves as a guide for the phylogenetic placement of CCRA specimens from Guam.

BLAST searches of the 10 putative Peyssonneliales species did not resolve their taxonomy either. Therefore, the COI-5P sequences of 32 Peyssonneliales specimens from this study were aligned with the 11 GenBank sequences (S3 Table) and *Bonnemaisonia asparagoides* (Woodward) C. Agardh as the outgroup. Priority was given to sequences of type species within a genus. If sequences of type species were unavailable, sequences of congeners were used. A phylogeny was generated from the alignment of the 32 Peyssonneliales specimens from the recruitment tiles plus the 11 reference sequences downloaded from GenBank, which

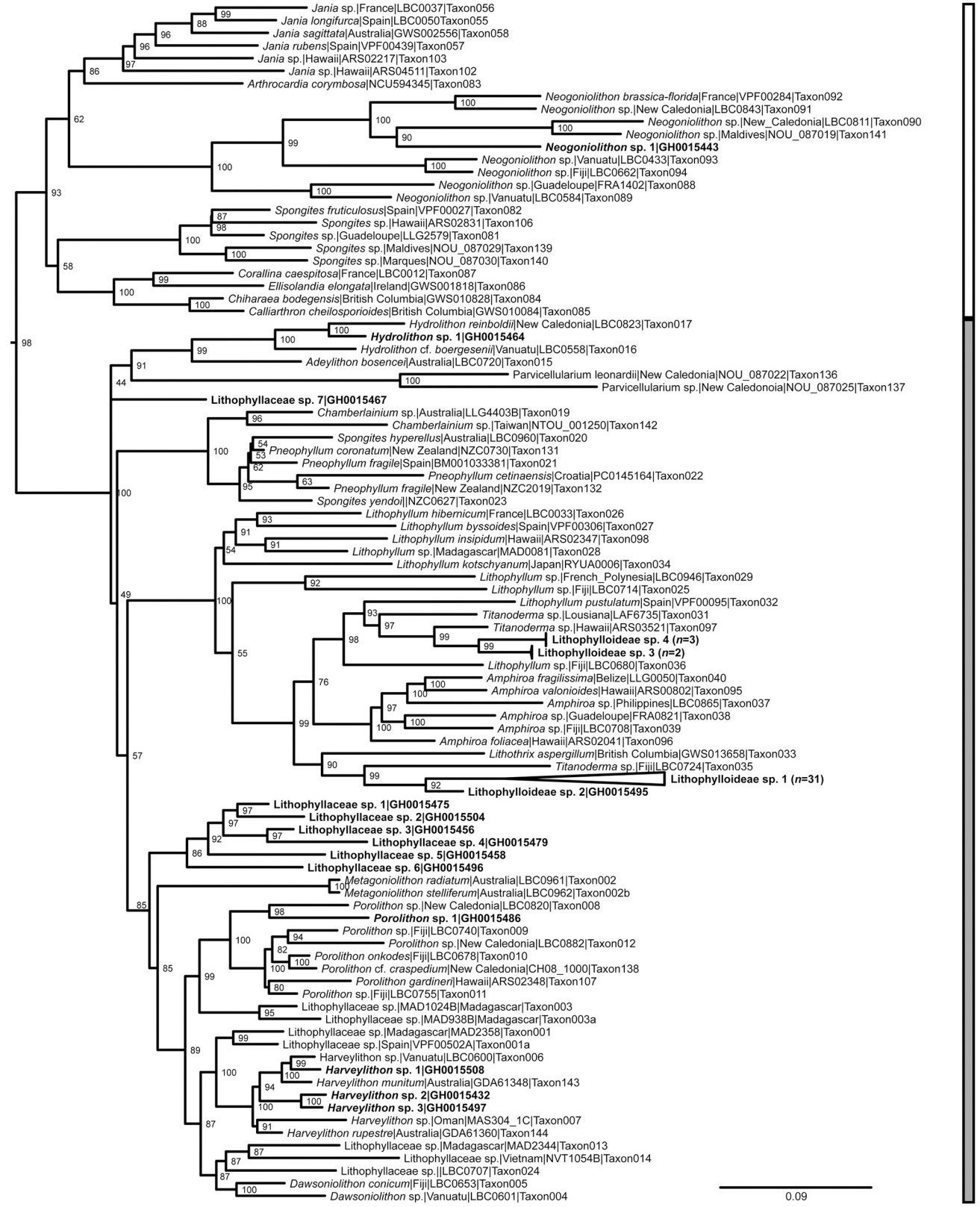

**Fig 2. Seven-gene concatenated maximum likelihood phylogeny for the 17 Corallinales species from the recruitment tiles (boldface font) and reference sequences from Peña et al. [68].** GH numbers are the specimens' voucher numbers in the Guam Herbarium (GUAM). The tree was inferred by maximum

likelihood with IQ-TREE [65]. Support values listed next to each node are ultrafast bootstrap approximations based on 1000 replicates. Scale bar represents nucleotide substitutions per site.

represented the following nine Peyssonneliales genera: *Incendia*, *Metapeyssonnelia*, *Peyssonnelia*, *Polystrata*, *Ramicrusta*, *Riquetophycus*, and *Sonderophycus*. The length of the COI-5P Peyssonneliales alignment was 621 bp. A maximum likelihood phylogeny was generated through IQ-TREE [66]. The best-fit evolutionary model was GTR+F+I+G4 (ModelFinder) [67].

## Statistical analysis

Data analysis was carried out in the R environment for statistical computing and visualization (v 3.5.1; R Development Core Team 2020). The percent cover of substrate categories on recruitment tiles was presented as mean ± standard deviation. A Kruskal-Wallis H test followed by a Tukey's Honest Significant Difference post-hoc test examined if the cover of CCRA substrate categories differed significantly from each other.

A G-test for goodness-of-fit (likelihood ratio or log-likelihood ratio) was used to evaluate whether *Acropora surculosa* larvae (1) were evenly distributed over the recruitment tiles and (2) preferred to recruit on specific substrate categories over others. This G-test was chosen because of the small sample size and the existence of one nominal variable with more than two values (substrate categories). The G-test evaluates if the observed number of coral recruits differs significantly from the expected number of recruits on individual tiles or on each of the substrate categories based on their percent cover. The null hypothesis for tile recruitment preference was that their occurrence was evenly spread over all the tiles. The null hypothesis for substrate recruitment preference was that the number of coral recruits per substrate category was a direct function of the percent cover of each substrate category, *i.e.*, random settlement.

## Results

### Species delimitation

Successful DNA sequences were obtained for 84 of the 89 algal specimens from the coral recruitment tiles, which were all representatives of the red algal class Florideophyceae. Of the 84 successfully sequenced specimens, 52 specimens belonged to the order Corallinales, while 32 specimens were representatives of the order Peyssonneliales. The 84 samples represented 27 distinct species (S1 Table). Of these 27 species, 17 species belonged to the order Corallinales and 10 were members of the order Peyssonneliales. Sequencing all three genetic markers for each putative species did not work reliably, but a minimum of two markers were obtained for 16 Corallinales species. No sequences from GenBank or BOLD matched (>97% sequence similarity) the 27 CCRA species from the coral recruitment tiles. Of the 27 putative species, one Corallinales species (Lithophyllaceae sp. 6) and one Peyssonneliales species (*Polystrata* sp. 3) matched the sequence of a CCRA that was previously collected from its natural habitat on Guam's reefs.

### Corallinales

Of the 17 Corallinales species, 16 species were identified as members of the family Lithophyllaceae with representatives for each of the four subfamilies: Lithophylloideae, Hydrolithoideae, Chamberlainoideae, and Metagoniolithoideae (Fig 2). One Corallinales species was identified as a representative of the family Neogoniolithoideae (Corallinaceae, Rhodophyta).

Four of the putative species in the Lithophyllaceae belong to the subfamily Lithophylloideae (Fig 2). Lithophylloideae sp. 3 and Lithophylloideae sp. 4 are sister taxa to *Lithophyllum*

*pustulatum* (J.V.Lamouroux) Nägeli and two other *Titanoderma* spp. Lithophylloideae sp. 1 and Lithophylloideae sp. 2 are members of a genus that warrants description. Lithophylloideae sp. 1 was represented by the most sequences in this study. Further phylogenetic analyses are required to resolve the taxonomy of these species and they were provisionally named as distinct species within the subfamily.

Six Corallinales species were confidently assigned to genus level. One species was well-resolved within the genus *Hydrolithon* (Hydrolithoideae, Rhodophyta; Fig 2). Four CCRA species belonged to the subfamily Metagoniolithoideae (Fig 2): *Harveylithon* sp. 1, *Harveylithon* sp. 2, *Harveylithon* sp. 3, and *Porolithon* sp. 1. All four species grouped with high support with their congeners. *Harveylithon* sp. 2 and *Harveylithon* sp. 3 are sister taxa, while *Harveylithon* sp. 1 is sister to a *Harveylithon* sp. from Vanuatu [70] (Fig 2). *Neogoniolithon* sp. 1 was the only species in this study that represented the family Corallinaceae (Fig 2) and was well-resolved within the *Neogoniolithon* clade (Fig 2).

Six species, Lithophyllaceae spp. 1–6, formed a clade within the Lithophyllaceae and sister to the Metagoniolithoideae (Fig 2). Each of these Lithophyllaceae spp. was represented by just a single sample, with at least 2 successfully sequenced markers. The nearest clade to Lithophyllaceae sp. 7 was the subfamily Hydrolithoideae and the recently described genus *Parvicellularium* [70] (Fig 2). Lithophyllaceae sp. 7 was a singleton specimen with only a successful *psbA* sequence.

## Peyssonneliales

Species delimitation of the 32 COI-5P sequences from Peyssonneliales specimens identified 10 distinct species. Three of these species were representatives of the genus *Polystrata* and seven species belonged to two distinct clades that could not be assigned to a recognized genus within Peyssonneliales (Fig 3). Peyssonneliales spp. 1–7 did not match nor were closely related to any sequenced species through a BLAST search. These seven Peyssonneliales members are not part of the clade with the type species of *Peyssonnelia*, *Peyssonnelia squamaria* Gmelin. Peyssonneliales sp. 1 and Peyssonneliales sp. 2 form a strongly supported clade. Peyssonneliales spp. 3–7 were resolved in another clade, with strong support for the subclade consisting of Peyssonneliales spp. 3–6 (Fig 3). Three *Polystrata* species were identified on the coral recruitment tiles (Fig 3).

## Substrate composition and *Acropora surculosa* settlement preference

Cover and community composition of all twelve tiles was comparable and all tiles were dominated by CCRA that did not show significant signs of pigmentation loss or bleaching (Fig 5). Corallinales members constituted the highest percent cover on the tiles (>50%; Fig 4; S4 Table), followed by bare substrate (~25%), and Peyssonneliales constituted the lowest cover (<20%; Fig 4; S4 Table). The tiles contained more Corallinales species than Peyssonneliales species. DNA barcoding revealed that a species richness that was difficult to morphologically identify. The 27 CCRA species would have been categorized as eight CCRA species if this study had only utilized morphological observation. Except for Lithophylloideae sp. 1 and Peyssonneliales sp. 5, the remaining 25 CCRA species were difficult to discern from one another during the image analysis. The different Corallinales species were more difficult to discern morphologically than the Peyssonneliales species. All coral recruits were associated with CCRA. One coral recruit was associated with a Corallinales spp. that displayed pigmentation loss but was not bleached.

Tukey's post hoc test concluded that the CCRA community composition did not differ significantly across all 12 tiles (Fig 5). However, significant differences in the percent cover

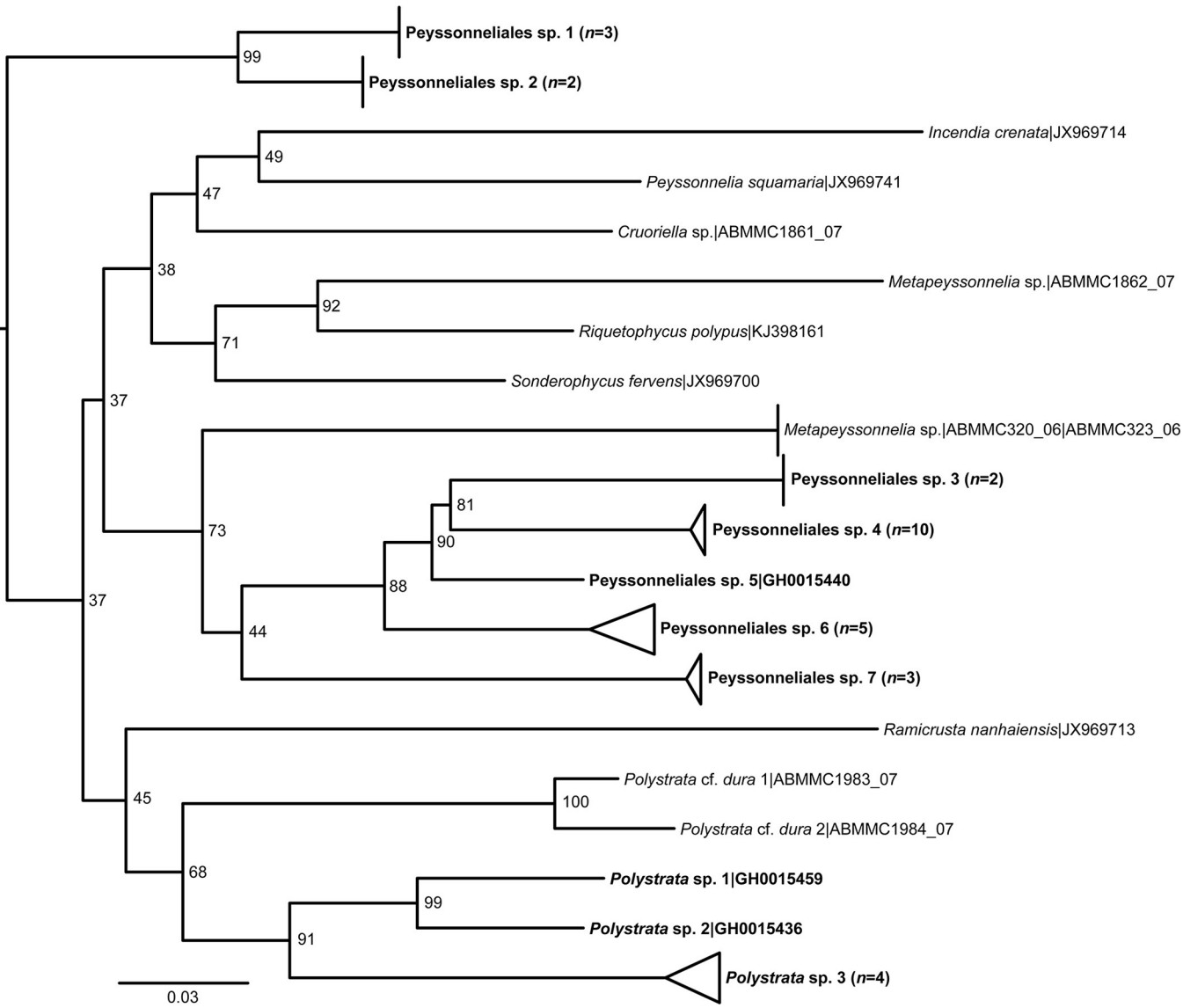

**Fig 3. Maximum likelihood phylogeny of the 10 Peyssonneliales species (boldface font) from the recruitment tiles and 10 reference sequences from GenBank inferred from COI-5P sequences.** The tree was inferred by maximum likelihood with IQ-TREE [65]. Support values listed next to each node are ultrafast bootstrap approximations based on 1000 replicates. Scale bar represents nucleotide substitutions per site.

between CCRA substrate categories was found. Lithophylloideae sp. 1 was the dominant species (41.5 +/- 8.5%) that covered the tiles ($p < 0.0001$; S4 Table). No significant preference in recruitment to any one of the 12 tiles was detected, but the G-test for goodness of fit showed that *A. surculosa* larvae favored certain substrate categories for recruitment ($P < 0.001$).

Lithophylloideae sp. 1 demonstrated to be an ecologically important CCRA species for *A. surculosa* recruitment, with an average cover of 41.5% and a total of 45 associated coral recruits (Fig 4; S4 Table). Lithophylloideae sp. 1 was also the only CCRA species in this study could visually be recognized, which was supported by 33 successfully sequenced specimens. covert Several of the sequenced Lithophylloideae sp. 1 samples had more than one coral recruit associated with them. The G-test clarified that the higher number of coral recruits on Lithophylloideae sp. 1 was more than just the result of the high percent cover of this species. *A.*

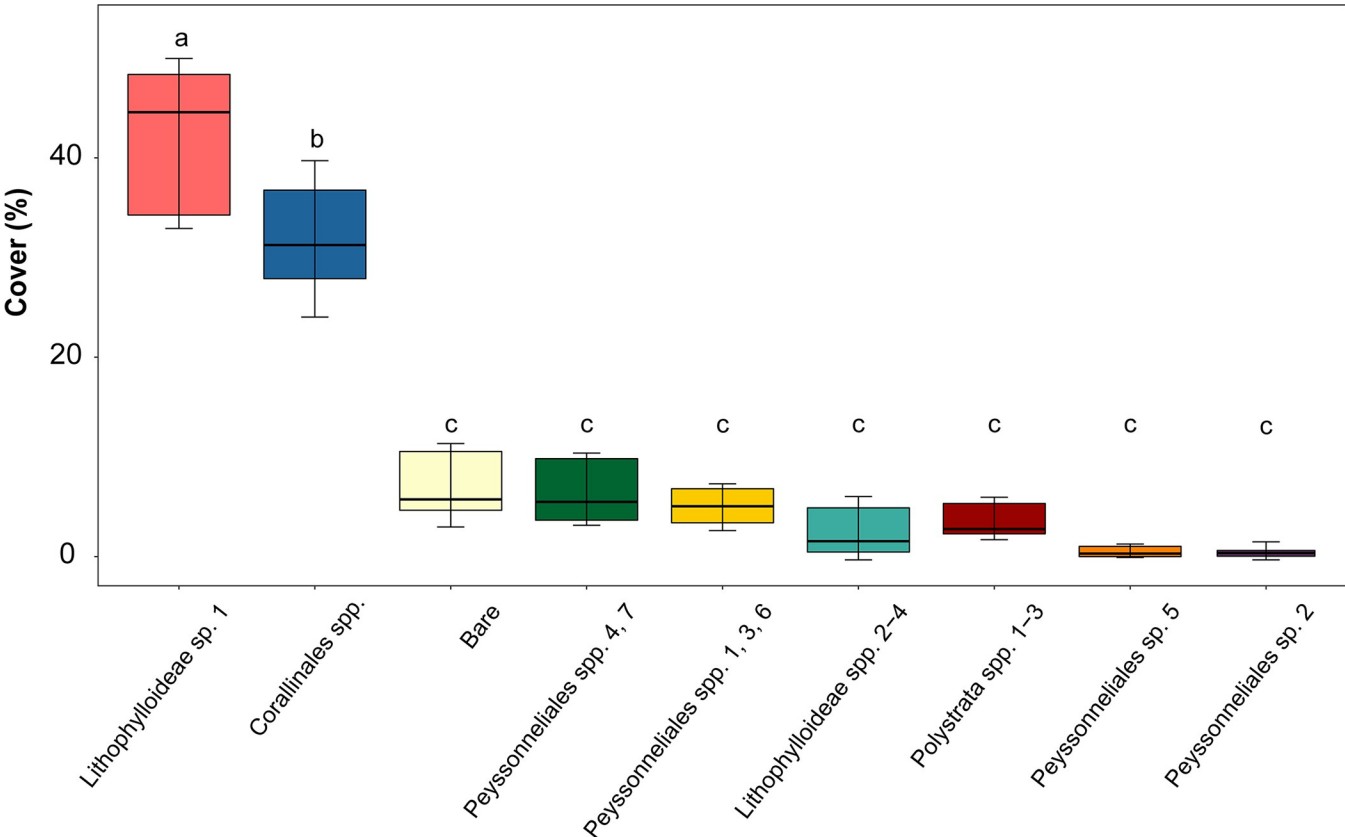

**Fig 4. Box plot of the percent cover for each substrate category on the 12 coral recruitment tiles.** The abscissa lists substrate categories, the ordinate shows percent cover. Median cover of a substrate category is indicated by a horizontal line in the interior of the box. The box represents the inter-quartile range (IQR) between the upper and lower quartiles. Whiskers indicate the minimum and maximum values beyond the IQR. Letters indicate significant differences between island groups ($P > 0.05$). Box colors match substrate category colors in Figs 1 and 5.

*surculosa* larvae significantly preferred this species as a recruitment substrate ($P < 0.0001$; S4 Table).

Lithophylloideae spp. 2–4 was the substrate category with the second most coral recruits (S4 Table). The average surface area of Lithophylloideae spp. 2–4 was 2.9% per tile (Fig 4). Despite its low cover, *A. surculosa* larvae also significantly preferred to recruit on this substrate category ($P = 0.037$; S4 Table). Of the seven remaining substrate categories, three had coral recruits on them: Corallinales spp., Peyssonneliales spp. 4,7 and *Polystrata* spp. 1–3. Corallinales spp. was the category that comprised the highest number of species (13 spp.; Table 1). Many of the species were represented by a single specimen. Corallinales spp. comprised a large percent cover on the recruitment tiles (31.9%; Fig 4) but the number of corals that recruited on this category was significantly lower than expected based on its percent cover ($P < 0.0001$; S4 Table). The remaining two categories with coral larval recruits were not statistically significantly preferred as recruitment substrates: Peyssonneliales spp. 4,7 had four recruits (6.8% cover; $P = 0.977$; S4 Table) and *Polystrata* spp. 1–3 had three recruits (3.8% cover; $P = 0.650$; S4 Table).

## Discussion

Successful dispersal and recruitment of invertebrate larvae are crucial to the health and recovery of tropical reef ecosystems following environmental disturbance [72]. Prior to this study, it

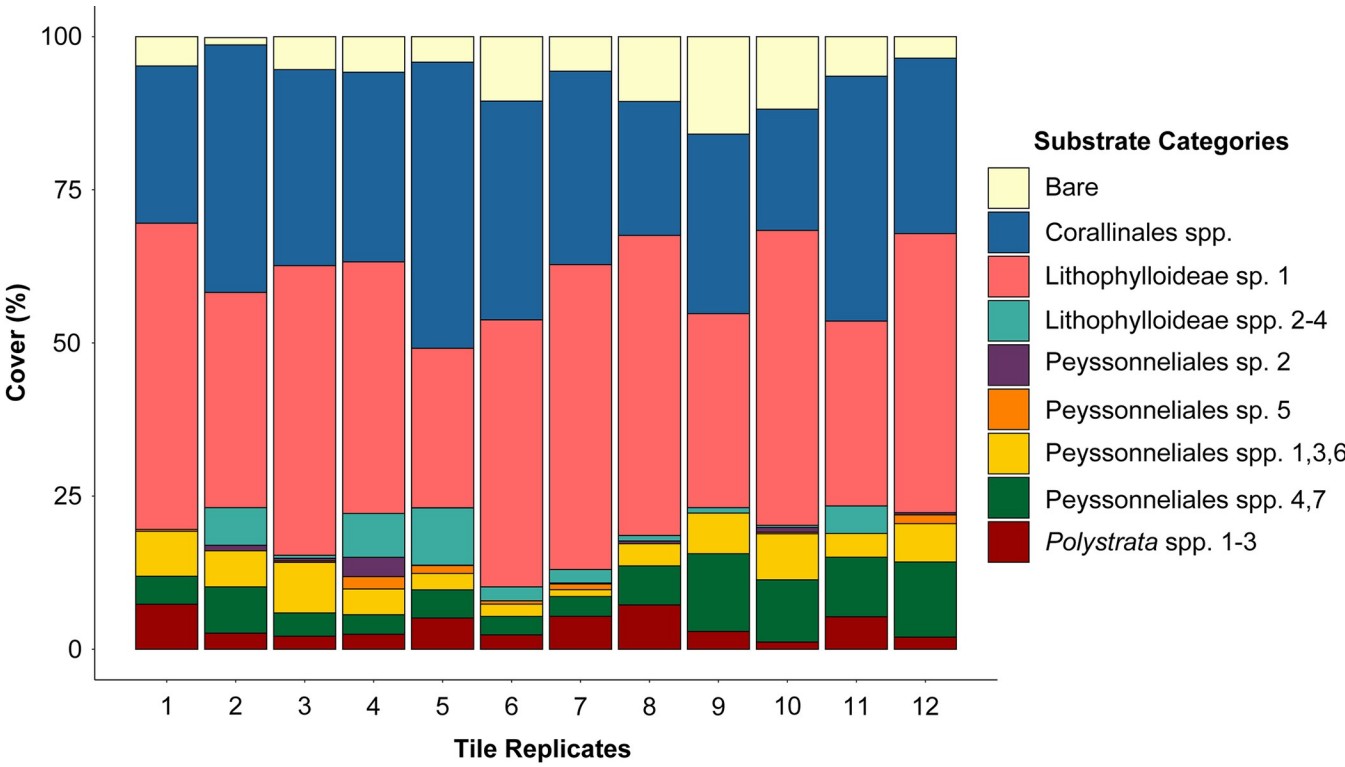

**Fig 5. Substrate community composition of the 12 coral recruitment tiles.** The abscissa lists the 12 recruitment tiles, the ordinate shows percent cover. Colors match substrate category colors in Figs 1 and 4.

was unclear which CCRA species served as the preferred recruitment substrates of *A. surculosa* in Guam. This study reports on (1) the diverse CCRA community composition of newly colonized bare substrates in a semi-natural environment and (2) the CCRA species that were significantly favored by *A. surculosa* recruits.

## Ecological significance of CCRA for *Acropora surculosa* recruitment

Despite their ability to recruit on a broad diversity of CCRA species, *Acropora* spp. have been reported to actively favor specific CCRA species as recruitment substrates when present [8, 9, 29, 30, 53]. Prior to this study, it was unknown if and which CCRA species were preferred by *A. surculosa* as recruitment substrates. Here we confirm that similar to other *Acropora* species, *A. surculosa* larvae actively favor certain CCRA species over others [8, 9, 29, 30, 52, 53]. The difference between this study and previous studies is the use of DNA barcoding and phylogenetic analyses to identify CCRA, whereas CCRA species identification in previous coral recruitment studies was mainly based on morphology [11, 28, 29, 30–35, 52, 53]. Using morphological identifications, *Lithophyllum prototypum* (Lithophylloideae, Corallinales; Foslie) has been reported to be a favored CCRA for coral larval recruitment throughout the tropics and subtropics [9, 29, 30, 53]. On the Great Barrier Reef, settlement of coral recruits on *L. prototypum* has been reported to be 15 times higher than a less preferred species, *Neogoniolithon fosliei* (Heydrich) Setchell & L.R.Mason [9]. Recruits of *Acropora tenuis* and *Acropora millepora* on *L. prototypum* also have the highest post-settlement survival rates [9]. Ritson et al. [53] observed *L. prototypum* having a lower benthic cover, yet *Acropora palmata* and *Montastraea faveolata* significantly preferred this species over more abundant CCRA in South Water Cay,

Belize. In an *in-situ* experimental study in Moorea, *L. prototypum* was the dominant of five CCRA species on coral recruitment tiles and was the most favored CCRA species for the recruitment and settlement of *Acropora* and *Pocillopora* larvae [29]. Despite the dominance of *L. prototypum* on these recruitment tiles, it was not a common CCRA on natural reef systems [9, 29, 30]. Since *L. prototypum* identifications in these studies were based on morphology [9, 29, 52, 53], they should be taken with reservation. *L. prototypum* and other taxa identified as *Titanoderma* spp. are regarded to be a minor component of Pacific reefs (low benthic cover) and have been found in the mid and outer zones of reef systems [29, 71]. Ongoing barcoding efforts of Guam's CCRA flora have not detected Lithophylloideae sp. 1, which was the preferred settlement substrata for *A. surculosa* in this study, on natural reefs yet. The lack of detection of Lithophylloideae sp. 1 in Guam's natural reef community is surprising due to its dominance in the CCRA community observed in this study.

Healthy communities of Lithophylloideae sp. 1 would benefit reef recovery following disturbance events that have caused coral die-offs. With growing evidence that many CCRA species have a high level of endemism and restricted distribution ranges [43–47], it is likely that the DNA barcoding of taxa previously identified as *L. prototypum* and *Titanoderma* spp. [9, 29, 30, 52, 53, 72] will reveal a diversity of cryptic species. Based on similarities in ecology and morphology, *L. prototypum* and Lithophylloideae sp. 1 could be closely related taxa of a currently undescribed genus. Further phylogenetic analyses in conjunction with morpho-anatomical observations are required to resolve the diversity and taxonomy of this ecologically important group of CCRA.

Lithophylloideae sp. 1 was morphologically distinct from the other 16 Corallinales species identified on the recruitment tiles. In visible light, it had a deeper reddish pink pigmentation with noticeably larger conceptacles compared to other CCRA species (Fig 1). Using fluorescence photography, Lithophylloideae sp. 1 displayed a fluorescent orange pigmentation that was not observed for the other 16 Corallinales species (Fig 1). This distinct fluorescence signature could result from unique pigments, chemical compounds or microbial communities that enhance successful acroporid recruitment. It has been proposed that successful coral larval recruitment and settlement are a function of the epiphytic microbiome communities and chemical signatures of CCRA [14, 30], which may be unique at the species level [35]. In studies where *L. prototypum* was strongly favored by *Acropora cytherea* larvae for settlement and attachment, the alga displayed a distinct microbial community and characteristic metabolomic fingerprint compared to other CCRA species [35, 72].

Despite the relatively high species richness of CCRA on recruitment tiles, only five CCRA species that were not Lithophylloideae members were associated with coral recruits. *A. surculosa* larvae were able to recruit onto Peyssonneliales sp. 4, *Polystrata* sp. 3, *Harveylithon* sp. 3, Lithophyllaceae sp. 4, and one Corallinales sp. Successful larval recruitment on non-favored CCRA species could be due to random settlement, but the many abiotic and biotic factors that contribute to larval settlement and recruitment are understudied. Each of the three Corallinales species with coral recruits (*Harveylithon* sp. 3, Lithophyllaceae sp. 4, and one Corallinales sp.) were only once recorded in the study and the true cover of these species could not be assessed. It is therefore difficult to assess their true suitability as coral recruitment substrates.

Ocean acidification can alter the chemical recognition between Corallinales species and coral larvae, which can lead to reduced coral recruitment [73]. Peyssonneliales species are believed to be less susceptible to ocean acidification than Corallinales species [74]. *A. surculosa* larvae recruited to two Peyssonneliales species (Peyssonneliales sp. 4 and *Polystrata* sp. 3). A study of coral larval recruitment by *Goniastrea retiformis* in Guam demonstrated that the larvae of this coral can recruit to Peyssonneliales species, but they are not the preferred recruitment substrates [28]. Studies in which *L. prototypum* was the favored recruitment substrate [9]

found that the post-settlement mortality on other CCRA was higher than on *L. prototypum.* Similar studies could be conducted to assess the long-term survival rate of *A. surculosa* colonies on other taxa, like Lithophylloideae sp. 1.

## CCRA species diversity and community composition

CCRA form a speciose and phylogenetically diverse group which is often treated as a single functional group in ecological and experimental studies [9, 29, 52, 53]. Studies using molecularly-identified species to discern the ecological composition of CCRA communities are less common [75]. Ecological studies of CCRA communities are commonly based on morphological observations, which lump CCRA species together [29, 52, 53, 76], which result in a severe underestimation of species diversity [46] The cryptic diversity revealed in this study emphasizes that CCRA species diversity is higher than what has been previously thought. By sampling thalli for DNA in conjunction with a photographic analysis, we were able to describe and quantify the CCRA community composition that was established on coral recruitment tiles. Further studies could resolve if the CCRA communities on the recruitment tiles in the experimental tank setup are unique in composition or if they also occur naturally in Pago Bay. The species richness of the CCRA flora in Pago Bay is of course much greater than those of the small recruitment tiles [77].

The increased use of DNA barcoding in floristic studies has allowed for more accurate and rapid diversity assessments than studies based on morpho-anatomical identifications [36, 56, 78–81]. For example, recently 72 new records of algal species, including members of the Corallinales and Peyssonneliales, were reported for four shallow reef sites in northern Madagascar [81]. Similarly, DNA barcoding documented 122 CCRA species from various sites around New Zealand and identified CCRA diversity hot spots around the island [46]. The most recent account of CCRA diversity for Guam was summarized in a checklist of Guam's seaweed flora [79], which lists 24 CCRA species for Guam based on morphological identification. Of these 24 species, 17 belong to the Corallinales, two belong to the Peyssonneliales, four belong to the Hapalidiales, and one belongs to the Sporolithales. Recently, two new species records and four new species of *Ramicrusta* (Peyssonneliales) were reported and described for Guam through molecular-assisted alpha taxonomy [56], increasing Guam's CCRA species count to 30. All of the 27 species on the recruitment tiles are representatives of the Corallinales and Peyssonneliales, resulting in a more than two-fold increase in the known species number of these orders for Guam [82]. It is unlikely that the 27 CCRA species identified in this study correspond to those reported in Lobban & Tsuda [82], as the previously reported taxa are quite distinctive and differ morphologically from the crusts in this study. Furthermore, an ongoing DNA barcoding study of Guam's CCRA flora reveals that many of the reported species are likely misidentifications. It is noteworthy that for each of the 13 non-Lithophylloideae Corallinales species only one specimen was sequenced, which complicated their morphological identification in the photographs of the community analysis. Without DNA barcoding, the number of CCRA taxa recognized on the tiles would have been reduced and a reliable assessment of the CCRA taxa that were favored by coral larvae might have been compromised [36]. These results suggest that cryptic diversity in CCRA around Guam is rampant, especially considering the small surface area of the tiles.

The small surface area from which the 92 CCRA samples were collected (106 cm$^2$ per tile) further supports the notion that CCRA species richness at small spatial scales can be hyperdiverse, even on substrates with little to no microhabitat diversity. The material, shape, and microfeatures of the recruitment tiles could have influenced the type of CCRA community that developed. Kennedy et al. [83] studied the recruitment and calcification of crustose coralline algae on six different experimental tiles (including ceramic, plastic, and glass tiles) on

three different habitats (backreef, forereef crest, and reef slope) at two orientations (horizontal and vertical) and found that community composition varied between tile material and orientation. Algal turf communities have also been characterized by high species richness at small spatial scales [84, 85]. Turf algal communities in Lhaviyani Atoll, Maldives, were found to be species rich and highly variable at microhabitat level (samples were separated by 10 cm) [86]. The CCRA communities of our recruitment tiles contained a similarly high species richness, but community composition was homogeneous between tiles. The latter finding is likely a function of the reduced environmental variability between experimental replicates. These tiles were all introduced into the same environment at the same time, resulting in synchronized CCRA recruitment and limited variation in CCRA community composition between tiles.

### Identity of CCRA taxa

Species delimitation based on DNA barcoding and phylogenetics has become standard practice in phycology, allowing for the rapid identification of (new) species [87]. While this approach has broadened our understanding of algal diversity, it has created challenges to describe and name algal species, including CCRA, since formal descriptions are time-consuming and trail behind the molecular identification of species [36, 40]. Of the 27 species identified, seven Peyssonneliales species and nine Corallinales species could not be reliably assigned to recognized genera. Upon completion of BLAST searches, these 16 species were not closely matched (>97%) with publicly available DNA sequences. Therefore, phylogenies were used to identify species to the lowest taxonomic level possible.

The validity of the genus *Titanoderma* has been debated [48, 88] and therefore the species from the recruitment tiles are referred to as Lithophylloideae spp. There are no reports of *Titanoderma* species in Guam or Micronesia. Lithophylloideae sp. 1 and Lithophylloideae sp. 2 are sister taxa to species named *Titanoderma* sp. and *Lithothrix* sp. in Peña et al. [69] (Fig 2). However, the type species *Titanoderma pustulatum* (J.V.Lamouroux) Nägeli, as *Lithophyllum pustulatum sensu* Peña et al. [69] (Fig 2), was resolved in a different clade. Our results suggest that the description of a new genus is warranted for the clade that contains Lithophylloideae spp. 1–2 and other taxa identified in the literature as *Titanoderma*. This clade is paraphyletic to the clade that contains *T. pustulatum* [42, 69, 89]. Further investigations into and a detailed description of Lithophylloideae spp. 1–2 is appropriate given the important ecological role of Lithophylloideae sp. 1 as a recruitment substrate for coral larvae.

### Conclusion

This study demonstrates that DNA barcoding is an effective tool for the detailed characterization of CCRA communities, which cannot be accomplished using morphological examinations. Given the high CCRA species richness on the small experimental tiles, it is expected that the floristic diversity of CCRA in the tropical Pacific is severely understudied. Experiments that investigate the ecological functions of CCRA are particularly valuable to assess and evaluate reef health in monitoring and environmental impact studies. Understanding that recruitment of acroporid larvae on CCRA is highly species-specific is important to guide coral reef management and conservation programs. The highly favored Lithophylloideae sp. 1 has yet to be found on natural reef systems. Further research on the diversity, community composition, and ecology of CCRA in tropical reef communities is required to build a holistic view of reef ecosystem functioning.

### Supporting information

**S1 Table. GenBank accession numbers for COI-5P, *psb*A, and *rbc*L sequences of specimens from the coral recruitment tiles, with indication of the order, taxon name, and herbarium**

**accession number.**
(DOCX)

**S2 Table. Taxon list from the Peña et al. [69] sequence alignment utilized to phylogenetically place each Corallinales species identified in this study.** The GenBank accession numbers is provided for each gene.
(DOCX)

**S3 Table. GenBank accession numbers or BOLD Systems identification numbers (\*) of COI-5P sequences of Peyssonneliales taxa used in Fig 3.** Bold taxa names are generitypes.
(DOCX)

**S4 Table. G-test for goodness of fit of recruitment preference to substrate categories.** Only substrate categories with coral recruits are listed. The first column lists the substrate categories with associated coral recruits. The second column shows the average precent cover of a substrate category on all tiles. The third column represents the total number of coral recruits associated with a substrate category. The last column shows the probability value of larval recruitment preference.
(DOCX)

## Acknowledgments

We are grateful to Dr. Laurie Raymundo and the Raymundo Coral Lab for the use of their coral recruitment tiles. The manuscript benefitted from conversations and assistance regarding R scripting and data analysis by Andrew McInnis. A special thanks to Dr. Alex Kerr and Dr. Heroen Verbruggen for their invaluable time in extending their knowledge. MD, MM, and TS are indebted to the University of Guam for supporting studies that document and conserve the natural resources of Guam and the greater Micronesian region.

## Author Contributions

**Conceptualization:** Mari Deinhart, Matthew S. Mills, Tom Schils.

**Data curation:** Mari Deinhart, Matthew S. Mills, Tom Schils.

**Formal analysis:** Mari Deinhart, Matthew S. Mills, Tom Schils.

**Funding acquisition:** Mari Deinhart, Matthew S. Mills, Tom Schils.

**Investigation:** Mari Deinhart, Matthew S. Mills, Tom Schils.

**Methodology:** Mari Deinhart, Matthew S. Mills, Tom Schils.

**Project administration:** Mari Deinhart, Matthew S. Mills, Tom Schils.

**Resources:** Mari Deinhart, Tom Schils.

**Software:** Mari Deinhart, Tom Schils.

**Supervision:** Mari Deinhart, Matthew S. Mills, Tom Schils.

**Validation:** Mari Deinhart, Matthew S. Mills, Tom Schils.

**Visualization:** Mari Deinhart, Matthew S. Mills, Tom Schils.

**Writing – original draft:** Mari Deinhart, Matthew S. Mills, Tom Schils.

**Writing – review & editing:** Matthew S. Mills, Tom Schils.

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
