## [Decision Letter · Decision Letter 0]

28 Sep 2021

PONE-D-21-27555Community assessment of crustose calcifying red algae as coral recruitment substratesPLOS ONE

Dear Dr. Deinhart,

Thank you for submitting your manuscript to PLOS ONE. After careful consideration, we feel that it has merit but does not fully meet PLOS ONE’s publication criteria as it currently stands. Therefore, we invite you to submit a revised version of the manuscript that addresses the points raised during the review process.

Myself and both reviewers find merit in your work. However, the reviewers also have extensive comments that span a range of issues that they identify. Please pay particular attention to the comments regarding the presentation of appropriate figures (e.g. suggestions to convert S1 into Fig 2).

We look forward to receiving your revised manuscript.

Kind regards,

Christopher Edward Cornwall, Ph.D.

Academic Editor

PLOS ONE

Journal Requirements:

Reviewers' comments:

Reviewer's Responses to Questions

**Comments to the Author**

1. Is the manuscript technically sound, and do the data support the conclusions?

Reviewer #1: Yes

Reviewer #2: Yes

2. Has the statistical analysis been performed appropriately and rigorously? 

Reviewer #1: Yes

Reviewer #2: I Don't Know

3. Have the authors made all data underlying the findings in their manuscript fully available?

Reviewer #1: Yes

Reviewer #2: Yes

4. Is the manuscript presented in an intelligible fashion and written in standard English?

Reviewer #1: Yes

Reviewer #2: Yes

5. Review Comments to the Author

Reviewer #1: Main Comments:

This is an interesting study that utilises modern molecular methods to examine diversity of crustose red algae and associated coral (Acropora surculosa) settlement patterns. However, there are parts of this manuscript that need significant revisions to improve the clarity, provide additional details on the methods and results of this study.

The use of these molecular methods to confirm identifications used in this study distinguishes this study from the increasing body of literature on coral settlement on coralline algae species. The large majority of previous studies are based on morphological identification which can be problematic due to the prevalence of cryptic species in coralline red algae. The authors should make a point of this throughout the manuscript.

Abstract – The second half of the abstract needs to be re-written as it is hard to follow and seems to jump back and forth between the taxonomy and coral settlement preferences. See specific comments below. It would also be a good to stress how this study is unique in the use of molecular methods to identify CCRA species.

Intro- The introduction is written well with clear rationale for the study and objectives.

Methods & results- The methods on the DNA barcoding and phylogenetic analysis are detailed with a few minor corrections needed. However, the information on the methods of coralline settlement and analysis of this data is severely lacking and it is unclear exactly how this was analysed. For example, there is no information on how successful coral settlement was detected and recorded and after how long this occurred. This first paragraph seems does not seem to be logically presented and could do with some re-ordering along with the additional information needed. The results from your statistical tests need to be presented somewhere in the manuscript, perhaps as tables in supplementary material. See specific comment below for more details.

Discussion- The discussion would benefit from some revision such as removing some sections that are not relevant to the discussion (e.g. the section regarding the anatomical diagnostic features of the genus Titanoderma). Additionally, the results of this study are compared to other studies in the pacific that show preferential settlement of coral species on T prototypum and it would be beneficial to point out that these were based on morphological studies that could likely be a mix of species lumped together. Furthermore, it would be nice to see further discussion around the diversity on the settlement tiles vs that of natural environments. Again, see specific comments below for more details.

In the future, please follow author guidelines closely and include page and line numbers on your manuscript to facilitate review. A copy of your manuscript with page and line numbers has been provided with this review for reference to my comments below.

Specific comments:

Abstract

Pg2 ln17 – What does “…crustose coralline algae, CCA) are often used at the functional group level…” mean? Please clarify. Does this mean that coralline algae species are often lumped together in many studies?

Pg2 ln20- You should state that molecular methods were used to identify the species in the aims section of the abstract as this is a distinguishing point of this research

Pg2 ln22 – You haven’t said what any of the coralline species are. Perhaps reword to “…preference of the coral Acropora surculosa to any identified CCRA species.”

Pg20 ln 24 – This sentence could be reduced to state that theses sequences do not match any in publicly available databases

Pg20 ln25-33 – This section of the abstract needs some rewording as it seems repetitive in parts and jumps back and forth between preferred settlement substrate and benthic cover on the tiles. I would avoid bring up the point that theses other pacific corallines are likely assigned to the wrong genus of Titanoderma as this confuses the reader and isn’t relevant to the point you are trying to make. Perhaps just state that your two Lithophylloideae species are closely related to other coralline species in the pacific that have been shown to be preferred coral recruitment substrate.

Introduction

Pg3 ln 45 – What do you mean by “…that deposit calcium carbonate.” Are you talking about in the cell walls or into the ocean/sediment

Pg3 ln 50 – What is a suppressor of nutrient indicator algae?

Pg5 ln 82-83 – Remove this final sentence of the paragraph. This isn’t relevant. If you want to retain this sentence you should be worded more broadly. i.e. A. surculose is a popular study organism for the following reasons.

Pg5 ln 97 – Perhaps cite Twist et al. 2020 as they discuss how caution must be taking with morpho-anatomical identification and the advantages of using molecular methods for coralline algae identification.

Twist, B. A., Cornwall, C. E., McCoy, S. J., Gabrielson, P. W., Martone, P. T., & Nelson, W. A. (2020). The need to employ reliable and reproducible species identifications in coralline algal research. Marine Ecology Progress Series, 654, 225-231

Pg6 ln 102 - Perhaps cite Hernandez-Kantun et al. 2016 and/or Maneveldt et al. 2017 as there is some evidence of wide species distributions with molecular identified corallines

Hernandez-Kantun, J. J., Gabrielson, P., Hughey, J. R., Pezzolesi, L., Rindi, F., Robinson, N. M., ... & Adey, W. (2016). Reassessment of branched Lithophyllum spp.(Corallinales, Rhodophyta) in the Caribbean Sea with global implications. Phycologia, 55(6), 619-639. sepecimens.

Maneveldt, G. W., Gabrielson, P. W., & Kangwe, J. (2017). Sporolithon indopacificum sp. nov.(Sporolithales, Rhodophyta) from tropical western Indian and western Pacific oceans: First report, confirmed by DNA sequence data, of a widely distributed species of Sporolithon. Phytotaxa, 326(2), 115-128.

Pg7 ln103 – Steneck and Diether [53] is probably an inappropriate reference for stating that current species richness in this group is underestimated.

Pg7 ln 104 – Reword to “CCRA are generally lumped together as a single function group in monitoring and ecological surveys...” or something similar

Pg7 ln 105 – Not just their role in tropical reefs but also in temperate reef systems and rhodolith beds.

Methods

Pg7 ln 121 – Please expand on the methods in this section to clearly describe how the coralline recruitment was recorded on the settlement plates. It is unclear how the tiles were covered in CCRA and if this was before the Coral settlement event or at the same time? How long were these tiles left for the CCRA to grow? How was settlement success of the coral recoded? Under a microscope? Using the photos? and after how many days? Is there a reason for using star shaped settlement tiles instead of circular or square ones?

Pg6 ln122-123 – This sentence doesn’t seem to be relevant or a good opening sentence for the methods. Perhaps start by explaining that star shaped tiles were left for X time to recruit CCRA and then state how A. surculose was settled in 2018. Then go onto explain how settlement success was recorded, that photos were taken for analysis and finally tissue sample was taken for DNA analysis for CCRA that A. surculose settled on along with 32 additional samples of distinct morphologies.

Pg7 ln124 – Is there a reference for SECORE protocols? No need to state that the Raymundo Coral lab follows these protocols.

Pg7 ln 133 – Were the tiles dried before DNA scrapping? If so, in air or silica gel?

Pg7 ln 146 – You don’t need to state that tissue samples were placed in 1.5ml epindorfs

Pg8 ln153 – COI isn’t the official marker for red algae. Saunders suggested that we utilize a common marker (COI) for red algae to make results comparable across studies, but it is not the "official" barcode marker.

Pg8 ln 158 – Please clarify what markers were used for what CCRA – Was COI one was used for everything (including Peyssonneliales) and psbA and rbcL were used for Corallinales, Sporolithales & Hapalidiales (although it appears only specimens of the order Corallinales were found).

Pg 8 ln 160 – Please include a citation for the statement that psbA is commonly used and has a high success rate. Perhaps Broom et al. 2008

Broom, J. E., Hart, D. R., Farr, T. J., Nelson, W. A., Neill, K. F., Harvey, A. S., & Woelkerling, W. J. (2008). Utility of psbA and nSSU for phylogenetic reconstruction in the Corallinales based on New Zealand taxa. Molecular phylogenetics and evolution, 46(3), 958-973.

Pg8 ln 165 – What do you mean by the amplification profile?

Pg9 ln172 – A BLAST search searches the Genbank database

Pg9 ln 180 – Please clarify the ABGD settings used and on what platform this analysis was run

Pg10 ln 200 – It should be clarified earlier in the methods that a separate analysis was run for the Peyssonneliales

Pg10 ln 214- I suggest moving this section on substrate cover up to underneath the first section in the methods or even integrated with this first section to improve the flow and clarity.

Pg11 ln 215-16 – What do you mean by substrate categories? Are these CCRA morphological/ putative species groups? And does this include things such as bare settlement plate. Please reword to clarify

Pg12 ln230 – Table 1 description- there is no “species count column”

Pg13 ln 245 – I do not understand why two separate Kruskal-Wallis tests were used. It seems like they were testing the same thing to see if there is a difference in substrate/CCRA cover across the tiles. Why isn’t this just one test?

Results

Pg14 ln 270 – Did any of the sequences match anything on the internal database (collected from natural reef systems in Gauam) that was mentioned in the abstract

Pg17 ln 291 – Why is figure 4 referenced here? Seems like it should be table 1

Pg15 Ln 293-298 – Some of this seems like discussion material. Perhaps just retain the first sentence about why you called them Lithophylloideae spp. and save the rest for the discussion

Pg17 ln 334 – Fig 3 caption – Your methods say you done a maximum likelihood tree, but the figure caption says it is a Bayesian tree. Which is it?

Pg17 ln339 – Make this heading bold font

Pg18 ln 365 – Please add the results of all statistical tests as a table in supplementary material

Pg19 ln. 366-67 – Please explain what you mean that the substrate categories fell into three groups based on their percentage cover. I don’t think you are trying to group substrate categories based on percentage cover, are you? You are just trying to determine if there is a difference in percentage cover between species.

Pg19 ln 371 – This sentence contradicts the one below. You say settlement doesn’t differ from random and then go onto say it does. The wording of the G-test results is confusing.

Pg19 ln 374-75 – You state that you ran multiple G tests in the results. Were these G-test p-values adjusted for multiple comparisons with something like a Bonferroni correction for multiple comparisons

Discussion

Pg20 ln404 - Perhaps adjust sentence to read “…substrates of A. surculosa in the western Pacific and if there was species specific settlement occurring.”

Pg21 ln418 – Can you expand on what you think the use of recruitment tiles had on the results of this study. Is there important CCRA species for coral settlement that may have been overlooked? Is it likely that CCRA from certain habitats settled on the tiles? Perhaps cite Kennedy et al. 2017 that looked at the use of different recruitment tiles in relation to natural communities.

Kennedy, E. V., Ordoñez, A., Lewis, B. E., & Diaz-Pulido, G. (2017). Comparison of recruitment tile materials for monitoring coralline algae responses to a changing climate. Marine Ecology Progress Series, 569, 129-144.

Pg21 ln421 – Was the study on Guam’s seaweed flora based on molecular identifications or morphology? If they were based on morphology alone, is it likely that some of your identified species match with these and further investigation of the type specimens is needed?

Pg21 ln428 – Change “doubling” to “double”

Pg22 ln433-34 – This is a good statement about how the diversity on the tiles would have been limited with morphological approaches alone. This could be expanded on and state that the if morphology alone had been used it would have affected the reliability of the coral settlement results

Pg22 ln445 – Would you consider your tiles to have high microhabitat diversity? Please expand

Pg22 ln446-456 – I am not sure what the point that is trying to be made in this paragraph. The first two sentence are a repeat of the results, and I am not sure what the middle two sentences mean (ln 450-454). Either remove this paragraph or reword to clarify the point you are trying to make.

Pg23 ln458 – perhaps rename the heading to “identity of CCRA taxa” as this is more resolving nomenclatural status and names than specie identification. I also think that this could be moved below the settlement section of the discussion.

Pg23 ln467 – You can’t recognise if a species belongs to a genus based on a BLAST search. You need phylogenetic trees to determine these relationships, which you have. Please re-word.

Pg23 ln472 – Is Titanoderma pustulatum in the phylogenetic tree (S1 fig)? I don’t see it. Please clarify that Titanoderma pustulatum is the type of the genus and therefore Lithophylloideae sp. 1 & 2 are not Titanoderma.

Pg24 ln475-488 - I would delete this whole last paragraph as it not relevant. It is important to resolve the identity of the species and apply appropriate names given their ecological importance. However, you did not measure any anatomical characters in this study, and I don’t think you need to get into a discussion about the validity of the genus Titanoderma and anatomical characters used to define it. What you have above this paragraph is sufficient saying more work is needed on the taxonomy.

Pg24 ln490 – Please provide citations for the statement in this first sentence. Also, state that this is for other Acropora species and not A. surculose

Pg24 ln496 – This would be a good place to bring up that these previous studies were probably based on morphological identifications that could have lumped multiple species under the name T. prototypum. Then point out that this is what distinguishes your study.

Pg25 ln515 – Do you mean species 1 not species 6?

Pg25 ln519 – I suggest starting this sentence with “For example, …”

Pg26 ln529 – How does competition explain successful larval recruit of coral on non-favoured CCRA? Please expand

Pg27 ln544 – This statement undersells your study. Maybe state that further investigation on post settlement survival could be beneficial for A. surculose.

Pg28 ln546-577 – This paragraph is a great conclusion!

Reviewer #2: This manuscript requires major revision. Terms need to be defined. The phylogenetic tree needs to be replaced with the one in supplementary material. The phylogenetic tree requires major revision. All of this explained in more detail on .docx with track changes. This is said to be a multigene tree, but for many taxa not all of the genes are provided. In some cases these genes are from DIFFERENT specimens, whose name may or may not be correctly applied. Only those sequences that can be linked to type sequences of corallines can be believed. Nearly all specimens identified based on morpho-anatomy are suspect. Nowhere is it indicated which sequences are linked to type specimens and which are not. All of this needs to be made clear to the reader, which currently is not done. Many sequences are identified only by a collection number and not by a herbarium accession number. This needs to be made clear to the reader in the Tables.

6. PLOS authors have the option to publish the peer review history of their article (what does this mean?). If published, this will include your full peer review and any attached files.

Reviewer #1: No

Reviewer #2: No

---

## [Author Response · Author response to Decision Letter 0]

31 Mar 2022

We thank the editor and the two reviewers for taking the time to provide generous comments on the manuscript, PONE-D-21-27555. A response to each point raised by the academic editor and reviewers was addressed. We hope the manuscript has been edited to satisfyingly address their concerns and the manuscript will now be suited for publication.

---

## [Decision Letter · Decision Letter 1]

6 Jun 2022

PONE-D-21-27555R1Community assessment of crustose calcifying red algae as coral recruitment substratesPLOS ONE

Dear Dr. Deinhart,

Thank you for submitting your revised manuscript to PLOS ONE; I sincerely apologise for the unusually delayed review timeframe. After careful consideration, we feel that it has merit but does not fully meet PLOS ONE’s publication criteria as it currently stands. Therefore, we invite you to submit a revised version of the manuscript that addresses the points raised during the review process. Your revised manuscript was assessed by both original reviewers, whose comments are appended below (more detailed feedback is also attached). You will see that both reviewers comment positively on the strength of the revisions. However, both reviewers raise minor concerns that should be addressed by appropriate revisions to the text.

Again, I sincerely apologise for the delay in communicating this decision. We look forward to receiving your revised manuscript.

Kind regards,

Emily Chenette

Editor in Chief

PLOS ONE

Journal Requirements:

Reviewers' comments:

Reviewer's Responses to Questions

**Comments to the Author**

1. If the authors have adequately addressed your comments raised in a previous round of review and you feel that this manuscript is now acceptable for publication, you may indicate that here to bypass the “Comments to the Author” section, enter your conflict of interest statement in the “Confidential to Editor” section, and submit your "Accept" recommendation.

Reviewer #1: (No Response)

Reviewer #2: (No Response)

2. Is the manuscript technically sound, and do the data support the conclusions?

Reviewer #1: Yes

Reviewer #2: Yes

3. Has the statistical analysis been performed appropriately and rigorously? 

Reviewer #1: Yes

Reviewer #2: Yes

4. Have the authors made all data underlying the findings in their manuscript fully available?

Reviewer #1: Yes

Reviewer #2: Yes

5. Is the manuscript presented in an intelligible fashion and written in standard English?

Reviewer #1: Yes

Reviewer #2: Yes

6. Review Comments to the Author

Reviewer #1: The authors have made great progress on this manuscript and improved the clarity, significance and addressed most of the issues raised by reviewers. However, some minor revisions are still needed on this manuscript. All the suggestions are explained in detail on an attached word document with track changes. The most significant area that needs work is the abstract, particularly the second half of the abstract where you talk about Titanoderma. There is no reference to how Titanoderma is related to any off the results you observed in the abstract and needs to be re-written. There are also several paragraphs throughout that are repetitive and/or not that relevant to the story being told. These paragraphs could be deleted, shortened or merged into other paragraphs.

Reviewer #2: Great job responding to the reviewers' suggested edits. The manuscript is significantly improved and the authors are to be commended for their detailed responses to reviewers' comments. On the attached are a few minor comments and edits.

7. PLOS authors have the option to publish the peer review history of their article (what does this mean?). If published, this will include your full peer review and any attached files.

Reviewer #1: No

Reviewer #2: No

---

## [Author Response · Author response to Decision Letter 1]

21 Jun 2022

Reviewer 1: The authors have made great progress on this manuscript and improved the clarity, significance and addressed most of the issues raised by reviewers. However, some minor revisions are still needed on this manuscript. All the suggestions are explained in detail on an attached word document with track changes. The most significant area that needs work is the abstract, particularly the second half of the abstract where you talk about Titanoderma. There is no reference to how Titanoderma is related to any off the results you observed in the abstract and needs to be re-written. There are also several paragraphs throughout that are repetitive and/or not that relevant to the story being told. These paragraphs could be deleted, shortened or merged into other paragraphs.

Thank you for taking the time to read our revisions and provide feedback to further improve this manuscript. The abstract, specifically the second half, has been edited per your suggestions (lines 15-39). We agree that the part discussing Titanoderma in the abstract did not fit with the message this manuscript wants to present so it was deleted. Repetitive paragraphs were removed from this revision, while other were merged into other paragraphs.

Reviewer 2: Great job responding to the reviewers' suggested edits. The manuscript is significantly improved and the authors are to be commended for their detailed responses to reviewers' comments. On the attached are a few minor comments and edits.

Thank you for taking the time to read through our manuscript and give suggestions to help improve it. We have incorporated you edits and suggestion.

---

## [Editor Report · Decision Letter 2]

1 Jul 2022

Community assessment of crustose calcifying red algae as coral recruitment substrates

PONE-D-21-27555R2

Dear Dr. Deinhart,

We’re pleased to inform you that your manuscript has been judged scientifically suitable for publication and will be formally accepted for publication once it meets all outstanding technical requirements.

Kind regards,

Emily Chenette

Editor in Chief

PLOS ONE
---

## [Editor Report · Acceptance letter]

13 Jul 2022

PONE-D-21-27555R2 

Community assessment of crustose calcifying red algae as coral recruitment substrates 

Dear Dr. Deinhart:

I'm pleased to inform you that your manuscript has been deemed suitable for publication in PLOS ONE. Congratulations! Your manuscript is now with our production department. 

Kind regards, 

on behalf of

Dr Emily Chenette 

Staff Editor

PLOS ONE